# Modelling factors in primary care quality improvement: a cross-sectional study of premature CHD mortality

Kate Honeyford,[1] Richard Baker,[1] M John G Bankart,[2] David Jones[1]

## ABSTRACT

**Objectives:** To identify features of primary care quality improvement associated with improved health outcomes using premature coronary heart disease (CHD) mortality as an example, and to determine impacts of different modelling approaches.

**Design:** Cross-sectional study of mortality rates in 229 general practices.

**Setting:** General practices from three East Midlands primary care trusts.

**Participants:** Patients registered to the practices above between April 2006 and March 2009.

**Main outcome measures:** Numbers of CHD deaths in those aged under 75 (premature mortality) and at all ages in each practice.

**Results:** Population characteristics and markers of quality of primary care were associated with variations in premature CHD mortality. Increasing levels of deprivation, percentages of practice populations on practice diabetes registers, white, over 65 and male were all associated with increasing levels of premature CHD mortality. Control of serum cholesterol levels in those with CHD and the percentage of patients recalling access to their preferred general practitioner were both associated with decreased levels of premature CHD mortality. Similar results were found for all-age mortality. A combined measure of quality of primary care for CHD comprising 12 quality outcomes framework indicators was associated with decreases in both all-age and premature CHD mortality. The selected models suggest that practices in less deprived areas may have up to 20% lower premature CHD mortality than those with median deprivation and that improvement in the CHD care quality from 83% (lower quartile) to 86% (median) could reduce premature CHD mortality by 3.6%. Different modelling approaches yielded qualitatively similar results.

**Conclusions:** High-quality primary care, including aspects of access to and continuity of care, detection and management, appears to be associated with reducing CHD mortality. The impact on premature CHD mortality is greater than on all-age CHD mortality. Determining the most useful measures of quality of primary care needs further consideration.

[1]Department of Health Sciences, University of Leicester, Leicester, UK
[2]Insitute of Primary Care and Health Sciences, Keele University, Keele, UK

**Correspondence to**
Kate Honeyford;
ceh28@le.ac.uk

## ARTICLE SUMMARY

### Strengths and limitations of this study

- Premature, rather than all age, coronary heart disease (CHD) mortality is the focus of the study; associations with several modifiable risk factors are considered and compared to those for all age CHD mortality.
- The study examines the impact of model choice and the measure of primary care quality on results and interpretation.
- This study identifies features of primary care, including relational continuity, associated with lower levels of premature CHD.
- The lack of reliable data relating to lifestyle factors such as smoking and obesity rates means that important covariates are not included in the model.
- Important predictors included in this study describe the adult population and are not specific to those under 75.

are increasingly being used to plan and manage health services.[1] A framework for health outcomes has been introduced to provide an overview of how the national health service (NHS) is performing and as a catalyst for driving quality.[2] One of the framework's five domains is the prevention of premature death, 1 of the 12 indicators in this domain being death from cardiovascular disease under the age of 75 years.

While mortality caused by coronary heart disease (CHD) has been falling steadily since the 1980s,[3] it is still the leading cause of years of life lost in the UK. The Global Burden of Disease Study[4] shows that CHD mortality rates in the UK are significantly above the mean of those in comparator countries. Although population characteristics including lifestyle (eg, obesity and smoking) and hypertension are known to be associated with CHD mortality, primary care may have a role in reducing CHD mortality. For example, in the USA, a greater supply of primary care physicians is associated with lower heart disease mortality,[5] although there is little evidence of a similar

## INTRODUCTION

In England, in common with health systems in many other countries, health outcomes

association in the UK.[6] Since general practices serve lists of registered patients, there is potential to relate mortality in practice populations to population characteristics and performance of practices, and thereby assist practices in planning and monitoring their activities to reduce population mortality.[7] A conceptual model describing how the delivery of primary care can modify the impact of population-based characteristics on health outcomes has been proposed.[8] It includes interventions which target the morbid and healthy populations through early detection, prevention and appropriate management of people with established disease and highlights the importance of access and continuity of care. However, the size of practice populations and the small numbers of deaths present methodological difficulties.[9] In seeking to understand how primary care performance affects mortality in practice populations, consideration must be given to (1) the measurement of the performance of primary care and (2) selection of the underlying statistical model.

In recent years, information about the performance of general practices has become available, in addition to other practice characteristics such as numbers of registered patients and numbers of general practitioners (GPs). The quality outcomes framework (QOF) has provided data in four domains (clinical; organisational; patient experience and additional). Within the clinical domain are several chronic conditions (including CHD), each made up of individual indicators. For each indicator, points are awarded to practices based on the percentage of patients for whom the target has been achieved (known as the underlying achievement). Score and underlying achievement for each indicator are publicly available. Additionally, QOF contains 'prevalence registers' showing the percentage of the practice population identified by the practice as having a particular condition or disease.

In investigating aspects of practice performance associated with various outcomes, Kiran et al[10] used combinations of individual indicators' underlying achievement to give a devised 'CHD quality achievement score'.

QOF prevalence registers have also been used as measures of quality of primary care; for example, greater numbers of people on practice hypertension registers in primary care trusts (PCTs) have been shown to be associated with lower CHD mortality, indicating that improved detection of hypertension has a positive impact on outcome.[8 11]

Differences between study results may be partly explained by different usage of QOF data. For example, Kiran et al[10] found that the overall CHD quality achievement score was negatively associated with CHD mortality. However, other studies have shown no association between *individual* indicators and CHD mortality[8] and CHD admissions[12] or overall scores and emergency and elective admissions.[13 14]

## Statistical method and model selection

Different statistical models may be used. A common method is multiple linear regression with age-standardised mortality rates as the dependent variable. However, since in research of this type age-adjusted variables are rarely available, multiple linear regression of standardised mortality rates can lead to biased risk estimates.[15–17]

Poisson models have the advantage of considering deaths as count data; age and sex can then be included in the model as explanatory variables, thus overcoming the lack of age-standardised explanatory variables. As the counts are often over-dispersed, a negative binomial model may be more appropriate than a Poisson model.[18] Models based on count data, for example, the numbers of deaths or hospital admissions, can also be found in the literature.[8 13 14 19 20]

## Premature mortality

This study defines premature mortality in those under 75, in line with the Outcomes Framework and Office of National Statistics preferences.[21] Few studies have considered the associations between quality of primary healthcare, as measured by QOF, and *premature* mortality. Allender et al[22] demonstrated a stronger association between socioeconomic deprivation and premature CHD mortality, compared to all-age CHD mortality. In a study of CHD hospitalisation and primary care,[12] patients were divided into the age groups 45–74 and 75 and over; socioeconomic status was found to be more important in the younger age group, but no difference was found in the association of quality of care and health outcomes in the two different age groups.

In this study, we investigate the choice of performance indicators and statistical models to explain premature and all-age practice population mortality from CHD. The overall aim is to identify key aspects of primary care quality improvement having potential to improve health outcomes here to reduce premature CHD mortality.

## METHODS

This is an observational study of CHD mortality between April 2006 and March 2009 in 229 practices in the East Midlands. All 230 practices that were open for 3 years of the study were eligible for inclusion. One practice offering a service to a restricted patient group was excluded from the study. The total population covered by the study is just under 1.7 million people. In this study, premature CHD mortality will be the main outcome measure; this will be compared with all-age CHD mortality. Counts of deaths will be modelled using negative binomial regression. Results will be compared to those from weighted linear regression of indirectly standardised mortality ratios to determine the impact of the model adopted on the overall findings and conclusions of the analysis.

CHD mortality counts for each practice were constructed from the Primary Care Mortality Database for the period, supplied by the relevant PCTs. Each record included the date of birth, date of death, underlying cause of death and the general practice code for the

patient's practice. Each death was linked to a general practice using these codes. Deaths with the underlying cause identified as CHD (International Classification of Diseases (ICD)-10 codes: I20–I25) were included in the study. In addition, indirectly standardised mortality ratios (SMRs) were calculated for each practice, based on England and Wales' standard rates and mid-year estimates of practice populations provided by PCTs in line with National Clinical and Health Outcomes Development (NCHOD)'s methodology.[23] Since the numbers of deaths in each practice per year were low, both counts and SMRs were aggregated over the 3 years.

QOF clinical data are publicly available for all practices in the sample for 2006/2007, 2007/2008 and 2008/2009.[24] The 2006/2007 indicators were used since the primary care described by indicators for that year will have had an impact on the largest proportion of patients included in the analysis.

Levene et al[8] classify primary healthcare into prevention, early detection and appropriate management. QOF indicators can be classified following this approach.

## Prevention

*Prevention of smoking and obesity*: Reducing the number of people who are smoking or are obese would be useful indicators of prevention; however, neither of these is available for practice populations. The QOF indicator, which details the percentage of patients who have a range of conditions and have been offered smoking cessation advice, is being used as a measure of prevention of smoking (SM02).

## Early detection

*Detection of hypertension*: Levene et al[8 11] argue that given that only a little more than half of the people found to have hypertension in population surveys in England are included on QOF hypertension registers, these can be used as a useful measure of hypertension detection.

## Appropriate disease management

Indicators identified by Levene et al have been selected for this analysis: (1) the control of serum cholesterol in patients on practice CHD registers (CHD08) and (2) the percentage of patients on the practice CHD register who are being treated with aspirin (or equivalent) (CHD09).

See table 1 for more detailed descriptions of QOF indicators.

The CHD quality achievement score devised by Kiran et al[10] is included in a comparative analysis. The measure is the mean underlying achievement of 12 indicators (see table 1).

## Access and sustained relationships

One question from the General Practice Patient Survey[25] relating to patients' recall of being able to consult a particular GP has been selected as an indicator of the ability to access care and whether patients prefer to have access to a sustained relationship in line with the work of Levene et al.[8] In addition, the number of GPs/1000 patients is being included as a measure of GP supply.

Characteristics of the practice population have been selected based on previous research, the use of count data and availability. The percentage of the practice population who are male and are aged over 65 are included because they are known to be associated with higher rates of CHD mortality. List size is included as a measure of exposure in the negative binomial model, to take into account variability in practice size, and therefore the number at risk varies from practice to practice. List size is included in multiple regression for consistency. Diabetes prevalence, ethnicity and deprivation are included as previous research has shown these to be important in explaining variation in CHD mortality (refs. 8 11 10 and 8 10, respectively). Neither obesity nor smoking rates are included as the medium super output area (MSOA) measures available are unlikely to describe rates within practice populations accurately. The sources of information used for each of these are summarised in table 2. Since CHD prevalence is highly correlated with age and hypertension detection, it was not included in the model to avoid collinearity problems.

Eight practices have missing data for the number of GPs (fte); three practices have missing data from the GP Patient survey. The main analysis was repeated using a range of values; choice of value was not important. Either the median value or values from the following years have been used.

## Statistical analysis

Counts of deaths in each practice are modelled here using negative binomial regression in preference to Poisson regression since the data are over-dispersed (over-dispersion parameter $\alpha=0.023$, $p<0.05$). To explore how sensitive the results and interpretation are to the modelling approach used, ordinary least squares (OLS) linear regression with the standardised mortality ratio (SMR) as the dependent variable was also carried out, using the inverse of the variance of SMR as a weighting to account for uncertainty in the standardised mortality ratios, in line with the work of Kiran et al.[10]

## RESULTS

Table 3 shows the summary data of counts of death, mortality rates and characteristics of practice populations and primary care for the 229 practices, serving approximately 1.7 million patients in 2006/2007. Table 4 shows the estimates of incident rate ratios (IRRs) in the presented negative binomial model, including 95% CIs and p values for all explanatory variables; the impact of a one unit increase in each explanatory variable is also included to aid the interpretation of IRRs. As in many models in this field reporting pseudo-$R^2$, the adjusted

**Table 1** QOF indicators used to compile the overall CHD quality achievement score

| Clinical domain | | | Summary statistics in this study (median and interquartile range) |
|---|---|---|---|
| CHD | | | |
| | CHD06 | The percentage of patients with CHD in whom the last blood pressure reading (measured in the previous 15 months) is ≤150/90 mm Hg | 90 (85–93) |
| | CHD08 | The percentage of patients with CHD whose last measured total cholesterol level (measured in the previous 15 months) is ≤5 mmol/L | 82 (78–87) |
| | CHD09 | The percentage of patients with CHD with a record in the previous 15 months that aspirin, an alternative antiplatelet therapy, or an anticoagulant is being taken (unless a contraindication or adverse effects are recorded) | 95 (93–97) |
| | CHD10 | The percentage of patients with CHD who are currently being treated with a β-blocker (unless a contraindication or adverse effects are recorded) | 73 (66–81) |
| | CHD11 | The percentage of patients with a history of myocardial infarction (diagnosed after 1 April 2003) who are currently being treated with an ACE inhibitor or angiotensin II antagonist | 92 (88–97) |
| Stroke and TIA | | | |
| | STROKE06 | The percentage of patients with TIA or stroke in whom the last blood pressure reading (measured in the previous 15 months) is ≤150/90 mm Hg | 88 (82–92) |
| | STROKE08 | The percentage of patients with TIA or stroke whose last measured total cholesterol level (measured in the previous 15 months) is ≤5 mmol/L | 76 (70–83) |
| Hypertension | | | |
| | BP05 | The percentage of patients with hypertension in whom the last blood pressure reading (measured in the previous 9 months) is ≤150/90 mm Hg | 77 (73–83) |
| Diabetes mellitus | | | |
| | DM12 | The percentage of patients with diabetes in whom the last blood pressure reading is ≤145/85 mm Hg | 80 (74–85) |
| | DM17 | The percentage of patients with diabetes whose last measured total cholesterol level (measured in the previous 15 months) is ≤5 mmol/L | 83 (77–87) |
| | DM20 | The percentage of patients with diabetes in whom the last HbA1C is ≤7.5 (or equivalent test/reference range depending on laboratory) in the previous 15 months | 67 (61–73) |
| Smoking | | | |
| | SMOKE02 | The percentage of patients with any of any combination of the following condition: CHD, stroke or TIA, hypertension, diabetes, COPD or asthma, who smoke and whose notes contain a record that smoking cessation advice or referral to a specialist service, where available, has been offered within the previous 15 months | 94 (92–96) |
| Kiran CHD overall achievement score | | Mean of the above 12 indicators | 83 (80–86) |

CHD, coronary heart disease; COPD, chronic obstructive pulmonary disease; DM, diabetes mellitus; HbA1c, glycated haemoglobin; TIA, transient ischaemic attack.

pseudo-$R^2$ is low, although the predicted and observed counts are relatively close (Lin's concordance coefficient 0.86). There is no pattern in the residuals and removal of the identified outliers did not materially affect interpretation of the model.

Table 5 shows the potential impact that changes in population or service characteristics could have on premature CHD mortality, given the current model. For each explanatory variable, the change from either the upper or lower quartile to the median is used to show how decreases in the 'risk' in the practice population or increases in the quality of primary care could impact on mortality, if the model is adequate and the relationships causal.

**Table 2** Characteristics of practice populations

| Measure | Source of data |
|---|---|
| Deprivation indices | Based on the Index of Multiple Deprivation 2007 |
| Percentage of GP list on diabetes register | QOF prevalence register 2006/2007 |
| Percentage of White ethnicity | Based on hospital episode statistics |
| Percentage of population who are over 65 | Based on mid-year practice population estimates provided by the PCT |
| Percentage of population who are male | Based on mid-year practice population estimates provided by the PCT |
| List size | Based on mid-year practice population estimates provided by the PCT |

GP, general practitioner; PCT, primary care trusts; QOF, quality outcomes framework.

Sensitivity analysis explored the impact of various smoking and obesity indicators. Inclusion of QOF registers or modelled estimates for local area obesity prevalence does not materially change the coefficients; there is no evidence that the model is improved by so doing. Similar results are found when modelled estimates of smoking rates and QOF indicators relating to smoking are included. None of these indicators has been included in the final model.

## Population characteristics

The main predictors of variation in premature CHD mortality of the population characteristics included in the model were deprivation, age (percentage aged 65 and over), percentage on the diabetes register and the percentage male. Table 5 suggests that practices in less deprived areas may have up to about 20% lower premature CHD mortality counts than those with deprivation at the median.

## Service characteristics

Increases in levels of service in terms of the percentage of patients on the CHD register whose serum cholesterol is controlled (CHD08), and the percentage of patients who recalled being able to see their preferred GP, were associated with lower levels of premature CHD mortality, as was (less clearly) increased hypertension detection (as measured through QOF hypertension prevalence registers).

Similarly, there is a suggestion that increases in GPs/1000 patients and the percentage of patients on the CHD register being treated with aspirin (or an

**Table 3** Summary statistics for mortality counts, population characteristics and service characteristics

| | Summary statistics in this study median (IQR) |
|---|---|
| **Mortality data** | |
| Indirectly standardised CHD mortality | 95.9 (78.3, 119.9) |
| Indirectly standardised CHD mortality—under 75 | 92.4 (64.0, 135.6) |
| Crude rate per 1000 | 1.30 (0.95, 1.57) |
| Crude rate per 1000—under 75 | 0.43 (0.28, 0.58) |
| **Practice characteristics** | |
| Deprivation indices | 16.2 (10.0, 27.0) |
| Percentage of GP list on diabetes register | 3.8 (3.3, 4.4) |
| Percentage of White ethnicity | 89.9 (77.5, 94.1) |
| Percentage of population who are over 65 | 14.7 (12.1, 17.0) |
| Percentage of population who are male | 50.2 (49.5, 51.2) |
| List size | 6435 (3749, 10319) |
| **Service characteristics** | |
| GPs/1000 | 0.55 (0.47, 0.64) |
| Percentage of patients with recalled perception of being able to see preferred GP | 88 (80, 93) |
| Percentage of GP registered list on hypertension register | 12.3 (11.0, 14.7) |
| The percentage of patients with any or any combination of the following conditions: coronary heart disease, stroke or TIA, hypertension, diabetes, COPD or asthma who smoke and whose notes contain a record that smoking cessation advice or referral to a specialist service, where available, has been offered within the previous 15 months SM02 | 94 (92, 96) |
| The percentage of patients with CHD whose last measured total cholesterol level (measured in the previous 15 months) is ≤5 mmol/L CHD08 | 82 (78, 87) |
| The percentage of patients with CHD with a record in the previous 15 months that aspirin, an alternative antiplatelet therapy, or an anticoagulant is being taken (unless a contraindication or adverse effects are recorded) CHD09 | 95 (93, 97) |
| CHD overall achievement score[10] | 83 (80, 86) |

CHD, coronary heart disease; COPD, chronic obstructive pulmonary disease; GP, general practitioner; TIA, transient ischaemic attack.

**Table 4**  Estimated incident rate ratios, associated 95% CIs and associated p values for negative binomial regression for premature (U75) CHD mortality count

| Explanatory variable | IRR | 95% CI | p Value | Effect of a 1% increase in explanatory variable on percentage change in premature mortality (CI) | |
|---|---|---|---|---|---|
| Percentage of White patients | 1.008 | (1.003 to 1.012) | 0.002 | +0.8% | (0.3 to 1.2) |
| Deprivation score | 1.017 | (1.011 to 1.024) | <0.001 | +1.7% | (1.1 to 2.3) |
| Prevalence of diabetes in 2006/2007 | 1.114 | (1.028 to 1.208) | 0.009 | +11.4% | (2.8 to 20.8) |
| Percentage over 65 | 1.059 | (1.038 to 1.081) | <0.001 | +5.9% | (3.8 to 8.1) |
| Percentage of male patients | 1.067 | (1.038 to 1.103) | <0.001 | +6.7% | (3.8 to 10.3) |
| Number of GPs/1000 patients | 1.197 | (0.885 to 1.619) | 0.244 | +19.7% | (−11.5 to 61.9) |
| Hypertension detection in 2006/2007 | 0.978 | (0.950 to 1.007) | 0.133 | −2.2% | (−5.0 to 0.7) |
| Percentage of patients offered smoking cessation advice (SM02) | 1.002 | (0.993 to 1.011) | 0.712 | +0.2% | (−0.7 to 1.1) |
| Percentage of serum cholesterol (CHD08) | 0.991 | (0.981 to 1.000) | 0.044 | −0.9% | (−1.9 to 0.0) |
| Percentage of aspirin (CHD09) | 1.002 | (0.982 to 1.022) | 0.884 | +0.2% | (−1.8 to 2.2) |
| Percentage of patients with recalled perception of being able to see preferred GP | 0.994 | (0.989 to 1.00) | 0.036 | −0.6% | (−1.1 to 0.0) |

Also included are the effects on premature CHD mortality count of a unit increase in the value of the explanatory variables.
*One unit on scale for deprivation score.
CHD, coronary heart disease; GP, general practitioner.

**Table 5**  Impact on premature (U75) CHD mortality count of an improvement in primary care or a decrease in population burden to the median from the upper or lower quartile as appropriate, given the current model

| Explanatory variable | Description of change | Effect of an improvement in primary care or a decrease in population burden on percentage change in premature mortality (CI) |
|---|---|---|
| Percentage of White patients | Decrease from upper quartile to median —decrease of 5.2% in percentage of white patients | −4.16% (−6.24% to −1.56%) |
| Deprivation score | Decrease from upper quartile to median —decrease of 10.8 units on scale | −18.36% (−25.92% to −11.88%) |
| Prevalence of diabetes in 2006/2007 | Decrease from upper quartile to median —decrease of 0.6% in diabetes prevalence | −6.84% (−12.48% to −1.68%) |
| Percentage over 65 | Decrease from upper quartile to median —decrease of 2.3% in percentage over 65 | −13.57% (−18.63% to −8.74%) |
| Percentage of male patients | Decrease from upper quartile to median —decrease of 1.0% in percentage of male patients | −6.7% (−10.3% to −3.8%) |
| Number of GPs/1000 patients | Increase from lower quartile to median—increase of 0.8 GPs per 1000 patients | 15.76% (−9.20% to 49.52%) |
| Hypertension detection in 2006/2007 | Increase from lower quartile to median—increase of 2.3% in detection | −5.06% (−11.27% to 1.61%) |
| Percentage of patients offered smoking cessation advice (SM02) | Increase from lower quartile to median—increase of 2.06% offered advice | 0.41% (−1.44% to 2.27%) |
| Percentage of serum cholesterol (CHD08) | Increase from lower quartile to median—increase of 4.0% in achieving serum cholesterol target | −3.6% (−7.6% to 0.0%) |
| Percentage of aspirin (CHD09) | Increase from lower quartile to median—increase of 2.0% in aspirin treatment | 0.4% (−3.6% to 4.4%) |
| Percentage of patients with recalled perception of being able to see preferred GP | Increase from lower quartile to median—increase of 8.0% in patients recalling being able to see preferred GP | −4.8% (−8.8% to −0.00%) |

CHD, coronary heart disease; GP, general practitioner.

alternative) are both associated with higher premature mortality. Table 5 allows the interpretation of these key variables in the current model, and suggests that increasing the percentage of patients who recall being able to see their preferred GP from a lower performing practice (typically 80%) to a median performing practice (88%) may reduce the premature CHD mortality counts by 4.8%. Similarly, increasing the percentage of patients on the CHD register for whom the serum cholesterol is below 5 mmol/L from a low performing practice (78%) to a median performing practice (82%) may reduce premature CHD mortality counts by 3.6%.

### Effect of choice of measure of quality of primary care
Similar results are found when the Kiran CHD quality achievement score is used in the model instead of the two separate CHD quality indicators. In the subsequent model, an improvement from 83% (lower quartile) mean achievement in the Kiran overall CHD achievement score to 86% (median) reduces CHD mortality counts by 3.6% (see online supplementary appendix 1 for more information).

### Effect of model selection
When a weighted multiple linear regression model for standardised mortality ratio was used, the directions of associations were the same as those described above. However, the CIs for the β-coefficients for the prevalence of diabetes and the CHD quality achievement measures now include zero and hence interpretation of the results might be less clear-cut (see online supplementary appendix 2 for more information).

### Comparison with all age mortality
The results for all age CHD mortality are broadly similar to those for premature CHD mortality. Increases in white ethnicity, deprivation and diabetes prevalence and an increase in GPs/1000 patients are associated with increases in mortality, whereas improvements in the remaining service characteristics correspond to decreases in mortality counts. However, the CIs for the IRRs for all service characteristics include one impacting on the interpretation of the importance of these variables.

### DISCUSSION
### Statement of principal findings
Population characteristics and the quality of primary care were found to be associated with variations in premature CHD mortality. Increasing levels of deprivation, the percentage of the practice population who were on the practice diabetes register, who were white, over 65 and who were male were all associated with increasing levels of premature CHD mortality. Control of serum cholesterol levels in those with CHD and the percentage of patients who could recall being able to see their preferred GP were both associated with decreased levels of

premature CHD mortality. Similar results were found when all-age mortality was considered. The combined measure of the quality of primary care was associated with a decrease in all-age and premature CHD mortality. However, it is difficult to determine which individual indicators within this measure are key to reducing CHD mortality. The evidence that hypertension detection is associated with decreased CHD mortality is less clear than has been found by Levene et al[8] at the PCT level. Different modelling approaches yielded qualitatively similar results; however, a detailed interpretation of the results would be model dependent, particularly if statistical significance were rigidly applied as a criterion of importance.

### Strengths and weaknesses of this study
A key strength of this study is that it considers the association of features of primary care with *premature* CHD mortality, and while the overall relationships are similar to those found when all-age mortality is considered, some associations are stronger, for example, that with patients' recall of being able to see their preferred GP.

Most explanatory variables in this study describe the adult practice population and not those under 75. To develop our understanding of the different contributions made by different variables in explaining premature mortality, it would be helpful to have reliable data describing different age groups within practice populations.

This study allows useful consideration of the impact of the measure of primary care quality used. While the higher underlying achievement in overall CHD achievement score devised by Kiran is associated with decreased premature mortality, this does not allow policymakers or clinicians to determine which of the 12 indicators are most important. An examination of the individual indicators shows that increasing the percentage of patients on the CHD register whose serum cholesterol is below 5 mmol/L is associated with decreasing mortality, but it is unclear whether this is the most important indicator. Unfortunately, as patient level data relating to QOF indicators are not available for this study, it is not possible to determine whether combinations of indicators within individuals are key or if there are interactions between indicators and other characteristics of individuals, for example, ethnicity or gender.

Reliable information about smoking rates, alcohol consumption and obesity within practice populations is lacking, and having access to these data would be likely to improve the model fit. While various estimates are available, many of these are modelled estimates for geographical areas, not practice populations, based on levels of deprivation, ethnicity and age, which are anyway included in the models in this study. Integrated Household Study smoking data are available at the local area level, and could be used to estimate smoking prevalence in practice populations. However, it was not possible to match these to practice populations for this study. The introduction of new smoking indicators in

the 2012/2013 QOF asks practices to record the smoking status of those aged 15 years and over and to record an offer of support and treatment to those who smoke, which may start to give a fuller picture of smoking prevalence in practice populations. While a QOF smoking indicator relating to smoking cessation advice has been included in this study as a measure of prevention, the wording asks only for 'an offer of support and treatment', which may throw doubt on its validity.[26] White ethnicity percentage was included in the model; it is highly correlated ($R_p$:−0.993) with the percentage who are Asian. More refined ethnicity information was not available for this study.

This study allows the impact of the statistical method and model selection to be considered. While the directions of associations are generally not affected by the choice of model, the statistical significance of the results varies between models for several key variables. For example, neither the prevalence of diabetes nor the CHD quality achievement score would be considered to be significantly associated with premature mortality using a weighted-linear regression model for SMRs.

### Relation to other studies
Previous research has shown that primary care is important in improving health outcomes but that the precise aspects of primary care, which are most important, are not clear.[8 10 13 14 27] This study confirms that position, but explores other aspects of the problem.

An association between continuity of care and reduced mortality in older patients has been found in the USA,[28] and Bankart et al[14] have shown that in practices with higher mean rates of satisfaction at being able to consult a preferred GP, the emergency admissions are lower. These findings give further support to the importance of continuity of care in improving health outcomes.

Levene et al have found that hypertension detection was associated with reduced levels of CHD mortality in two studies completed at the PCT level. In their Global Burden of Disease Study 2010, Murray et al[4] highlight the importance of early detection and long-term management of high blood pressure as a 'clear route to accelerate progress for the leading causes of avoidable cardiovascular mortality'. However, the evidence of the importance of hypertension detection in this study is not clear-cut. It may be that QOF prevalence registers are not a useful measure of detection for individual practices. Low QOF prevalence rates may be due to lower levels of hypertension rather than lower detection, and conversely high QOF prevalence rates may be due to higher underlying prevalence rather than improved detection. More work to identify the most useful combination of QOF indicators to summarise hypertension detection is necessary.

Consistent with other studies, socioeconomic deprivation is the main predictor of CHD mortality. Bottle et al[12] and Allender et al[22] have found that socioeconomic deprivation is more strongly associated with mortality in younger age groups and this study has confirmed this pattern.

While some studies clearly state the reasons for their selection of the statistical modelling approach and/or potential explanatory variables (eg,[13 14 19 29 30]), many studies do not. Here we find some indication that qualitative interpretations of results are robust to model choice. However, a more detailed interpretation of results is likely to vary between models, particularly if there is undue reliance on simplistic interpretation of statistical significance.

### Meaning of the study
This study adds to the body of research demonstrating that high-quality primary care is associated with improving health outcomes. Aspects of continuity of care and disease management have been identified as having a bigger impact on reducing premature CHD mortality than all-age CHD mortality. While the most important individual indicators relating to disease management have not been identified, there is clear evidence that improving achievement in QOF indicators is associated with decreasing CHD mortality. The importance of continuity of care, again shown here, strongly suggests that this is an area for general practices to prioritise. The findings also suggest that data about outcomes such as premature CHD mortality could be used by practices to monitor and, over several years, plan their care to improve population health.

The ongoing importance of socioeconomic deprivation in explaining higher levels of mortality cannot be ignored. Understanding the relationship between deprivation and health outcomes more precisely remains an important area of further study.

### Unanswered questions and future research
The lack of reliable practice level information on key areas such as obesity, alcohol and smoking prevalence has an important impact on primary care research and is an important health information issue needing effective attention as the NHS undergoes major changes. While QOF indicators relating to smoking prevalence have been introduced, the reliability of these measures will need to be scrutinised.

To further our understanding of the relative importance of different QOF indicators, it would be useful to study individual QOF indicators at the patient level to see how they interact with each other and with the characteristics of individuals on an individual patient basis. In future studies, the impact of the introduction of health checks in 2009 should be explored.

### CONCLUSION
Improving the quality of primary care will play an important part in decreasing premature mortality, and there is evidence that high underlying achievements in

QOF clinical indicators are a useful measure of quality primary care. Continuity of care, in a country with universal access to healthcare, is important and should not be underestimated by policymakers and clinicians. Nonetheless, lifestyle factors are also important, but our ability to study them adequately in primary care, or to evaluate the role of primary care in addressing them, is currently limited by the quality of measures at the practice level. If primary care services, delivered by clinical commissioning groups are to be monitored and developed using the new NHS Outcomes Framework, better data and careful modelling and interpretation are vital.

**Contributors** The study was conceived by KH, RB, JB and DJ. KH designed the study, carried out the analysis and drafted the initial manuscript. JB and DJ contributed to the statistical analysis. RB, JB and DJ contributed to the drafting and editing of the final manuscript and also interpreted and reviewed the results of the statistical analysis.

**Funding** The research was funded and led by National Institute for Health Research (NIHR) Collaboration for Leadership in Applied Health Research and Care) based at LNR. The views expressed are those of the authors and not necessarily those of the NHS, the NIHR or the Department of Health. The funders had no role in the study design, data collection and analysis, decision to publish or preparation of the manuscript.

**Competing interests** KH had financial support from CLAHRC in the form of funding for PhD fees.

**Provenance and peer review** Not commissioned; externally peer reviewed.

**Data sharing statement** No additional data are available.

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
