## [Reviewer comments · BMJ Open]

Some articles will have been accepted based in part or entirely on reviews undertaken for other BMJ Group journals. These will be reproduced where possible.

ARTICLE DETAILS

TITLE (PROVISIONAL)	Modelling factors in primary care quality improvement: a cross-sectional study of premature CHD mortality.
AUTHORS	Honeyford, Kate; Baker, Richard; Bankart, M. John; Jones, David

VERSION 1 - REVIEW

REVIEWER	Michael Soljak Clinical Research Fellow Department of Primary Care & Public Health
REVIEW RETURNED	10-Aug-2013

THE STUDY	The research objectives/hypotheses should be stated more clearly. In terms of appropriateness of overall study design, a patient level analysis is the gold standard, to prevent the ecological fallacy. This analysis is not yet possible as patient level data from GP research databases anonymises practices, so the method is the best currently available. Leicester has a high South Asian population in which CHD prevalence is higher than Europeans, but this is not mentioned and could have been included in the models. Inexplicably CHD prevalence is also not included as a population factor. Smoking prevalence is referred to but is said to be unavailable. Practice level smoking estimates are can be obtained from the Integrated Household Survey (NB this is not a modelled estimate, it is a sample). Local practice level staffing data would also be available to the authors.
REPORTING & ETHICS	The conclusions would be more robust if S Asian population, CHD prevalence and smoking prevalence had been included.
GENERAL COMMENTS	Using premature mortality as an outcome is original and relevant to current health policy. The use of modelled effect estimates is an innovative way to explain effect sizes to a wider audience. However the conclusions would be more robust if other important population predictor variables could be included as suggested.

REVIEWER	Dr Kiran Patel, Cardiologist and Medical Director, Heart of England NHS trust, Birmingham UK
REVIEW RETURNED	27-Aug-2013

THE STUDY	The study essentially defines which aspects of a primary care quality improvement strategy improves outcomes using CHD as an example, rather than an intervention targeting premature CHD - perhaps the title and hypothesis should be rephrased as such. The impact of NHS Health Checks is not mentioned and this would
------------------	---

	have been apparent for the time period used in the study as a confounder. 1.State early on that the definition of premature CHD varies and clearly state the definition used in this study differs to that used in the NSF for CHD vs the Outcomes framework
RESULTS & CONCLUSIONS	The study essentially defines which aspects of a primary care quality improvement strategy improves outcomes using CHD as an example, rather than an intervention targeting premature CHD - perhaps the title and hypothesis should be rephrased as such. The impact of NHS Health Checks is not mentioned and this would have been apparent for the time period used in the study as a confounder.

VERSION 1 – AUTHOR RESPONSE

M Soljak

The research objectives/hypotheses should be stated more clearly.

>These have been modified in light of the second reviewer's comments (see title page (p10), abstract (p2) and introduction (p5)).

South Asian population

>The percentage of the practice population who are white is included in the study; this is highly correlated with the percentage who are Asian (PR=-0.993, p<0.01). A comment has been added to the discussion to note and explain this (p10).

CHD prevalence is also not included as a population factor.

>Neither Kiran nor Levene included CHD prevalence in their model. Whilst some researchers do so, it is not a universally adopted approach. We believe that the factors influencing both the prevalence and mortality are of more interest and importance. In addition, CHD prevalence is highly correlated with percentage over 65 and hypertension detection, and its inclusion would introduce issues relating to collinearity into the model. If we do include it in the model it is a non-significant predictor (IRR:0.985 95%CI:(0.867, 1.112) and has little impact on other estimates and conclusions. A note explaining this has been added to the methods section (p7).

Smoking prevalence

>It was not possible to match MSOA level data to practice populations for this study. Whilst IHS data can be used to estimate practice population smoking prevalence, the data are available only for the last year of the period of study and the process involves estimating from small numbers. A note has been included in the text regarding IHS data (p10).

>We agree that the conclusions would be more robust if appropriate measures of smoking prevalence could be included. We comment in the conclusion that there is still a need for reliable smoking prevalence data to be available at practice level if primary care services are to be monitored and developed (p13).

Local practice level staffing data would also be available to the authors

>GP staffing data were in fact included in the study (see methods p7).

The conclusions would be more robust if S Asian population, CHD prevalence and smoking prevalence had been included.

>The authors believe that percentage white ethnicity is a reasonable proxy variable for the

complement of Asian population practice percentage, and that CHD prevalence would not improve the model and its interpretation and would indeed introduce issues of collinearity. We agree that inclusion of an appropriate measure of smoking prevalence would improve the model and make the conclusions more robust, but have explained why such a measure is not available (p10).

Dr Kiran Patel

The study essentially defines which aspects of a primary care quality improvement strategy improves outcomes using CHD as an example, rather than an intervention targeting premature CHD - perhaps the title and hypothesis should be rephrased as such.

>The title (p1), aim (p2) and hypothesis (p5) have been modified in response to these comments.

The impact of NHS Health Checks is not mentioned and this would have been apparent for the time period used in the study as a confounder.

>NHS health checks were introduced in 2009/2010 and therefore were not included in the study, as we studied mortality up to March 2009 only. The impact of NHS Health Checks should be taken into account in future research relating to more recent periods (noted on p12).

State early on that the definition of premature CHD varies and clearly state the definition used in this study differs to that used in the NSF for CHD vs the Outcomes framework

>The definition of premature mortality used in this study has been emphasized (Abstract (p2) and p5). A debate about the most useful definition of premature mortality may be useful, but space is insufficient to include it in this paper.